# Environmental Management of *Legionella* in Domestic Water Systems: Consolidated and Innovative Approaches for Disinfection Methods and Risk Assessment

**DOI:** 10.3390/microorganisms9030577

**Published:** 2021-03-11

**Authors:** Emanuele Luigi Sciuto, Pasqualina Laganà, Simona Filice, Silvia Scalese, Sebania Libertino, Domenico Corso, Giuseppina Faro, Maria Anna Coniglio

**Affiliations:** 1Azienda Ospedaliero Universitaria Policlinico “G. Rodolico-San Marco”, Via Sofia 78, 95123 Catania, Italy; e.l.sciuto@gmail.com; 2Regional Reference Laboratory of Clinical and Environmental Surveillance of Legionellosis, Messina, Department of Biomedical and Dental Sciences and Morphofunctional Imaging, University of Messina, Torre Biologica 3p, AOU ‘G. Martino, Via C. Valeria, s.n.c., 98125 Messina, Italy; plagana@unime.it; 3Istituto per la Microelettronica e Microsistemi–Consiglio Nazionale delle Ricerche (CNR-IMM), Ottava Strada 5, 95121 Catania, Italy; simona.filice@imm.cnr.it (S.F.); silvia.scalese@imm.cnr.it (S.S.); sebania.libertino@imm.cnr.it (S.L.); domenico.corso@imm.cnr.it (D.C.); 4Azienda Sanitaria Provinciale di Catania, Via S. Maria La Grande 5, 95124 Catania, Italy; giuseppina.faro@aspct.it; 5Regional Reference Laboratory of Clinical and Environmental Surveillance of Legionellosis, Catania, Department of Medical and Surgical Sciences and Advanced Technologies “G.F. Ingrassia”, University of Catania, Via Sofia 87, 95123 Catania, Italy

**Keywords:** *Legionella*, water systems, Water Safety Plan, disinfection methods, emerging technologies

## Abstract

*Legionella* is able to remain in water as free-living planktonic bacteria or to grow within biofilms that adhere to the pipes. It is also able to enter amoebas or to switch into a viable but not culturable (VBNC) state, which contributes to its resistance to harsh conditions and hinders its detection in water. Factors regulating *Legionella* growth, such as environmental conditions, type and concentration of available organic and inorganic nutrients, presence of protozoa, spatial location of microorganisms, metal plumbing components, and associated corrosion products are important for *Legionella* survival and growth. Finally, water treatment and distribution conditions may affect each of these factors. A deeper comprehension of *Legionella* interactions in water distribution systems with the environmental conditions is needed for better control of the colonization. To this purpose, the implementation of water management plans is the main prevention measure against *Legionella*. A water management program requires coordination among building managers, health care providers, and Public Health professionals. The review reports a comprehensive view of the state of the art and the promising perspectives of both monitoring and disinfection methods against *Legionella* in water, focusing on the main current challenges concerning the Public Health sector.

## 1. Introduction

*Legionella* colonization of building potable water systems implies many Public Health concerns, especially for more fragile persons (elderly adults, smokers or people with weakened immune systems), which are particularly susceptible to the infection and at risk for developing clinical complications and respiratory failure. Obviously, as the population ages, the health impact on ‘at risk’ groups of legionellosis is likely to continue to increase.

Isolation and identification of *Legionella* from the environment is crucial for the management of environmental and clinical prevention, as well as for epidemiological purposes and for outbreak investigations. Culture methods are the gold standard for the detection of *Legionella* in environmental samples. Hazard analysis is also useful within the context of a control risk management plan. In addition, the adoption of chemical or physical disinfection methods is essential to contain the contamination.

This review focuses on historic, current, and emerging monitoring and disinfection methods against *Legionella*. State of the art and promising perspectives are reported. Moreover, challenges facing health care professionals, building managers, and the Public Health sector are discussed.

## 2. Methods

Search for literature was carried out using the keywords: ‘*Legionella*’ OR ‘Legionnaires’ Disease’ NEAR ‘prevention’ AND ‘water systems’ AND ‘disinfection methods’ NEAR ‘*Legionella*’ AND ‘risk assessment’ AND ‘Water Safety Plan’. The search included published papers between 1980 and 2020, and it was conducted in relevant biomedical and engineering databases: ACS Publications, Elsevier, JSTOR, PubMed, SDOS, and Wiley Online Library. US and European technical reports and guidelines for the prevention of legionellosis were also included. Inclusion criteria were studies about: (*i*) growth and survival of legionellae in water systems, (*ii*) effectiveness of conventional and emerging treatment technologies under in vitro conditions and after follow up in real conditions, (*iii*) water sampling and emerging detection methods, and (*iv*) water systems management programs.

The literature review was divided into four main parts to evaluate the following aspects of environmental management of *Legionella* in domestic water systems: (*i*) factors that regulate *Legionella* growth and survival, (*ii*) disinfection methods, (*iii*) monitoring methods, and (*iv*) risk assessment.

As reported in Figure 1, after the initial search through databases, n. 300 records were identified for further review. Additionally, n. 11 records (e.g., EU regulations, technical reports, guidelines, kit datasheets, etc.) were identified through other web sources. Once the duplicates were removed, a total of n. 98 records were excluded because they were out of the topic. Finally, among the full-text articles, n. 166 were confirmed because they matched the inclusion criteria.

Each article was evaluated with respect to the environmental factors related to the growth of *Legionella* in water systems, if the management of the water distribution system was associated with the control of *Legionella*, and if the management strategies were adequately tested for effectiveness and/or accuracy.

## 3. Factors Related to the Growth and Survival of Legionellae and the Management of Environmental Prevention

Legionellae are ubiquitous in natural and man-made water systems. A few, and especially *Legionella longbeacheae*, are also isolated from non-aquatic habitats such as potting soil and compost [1]. The correlation between *Legionella* colonization of a water system and the risk of acquiring the disease has been well established. When colonization occurs within a water system, abatement of *Legionella* is difficult and generally eradication is not possible because the bacterium has found the optimal environmental conditions and because it can activate surviving strategies.

Favorable conditions for the multiplication of *Legionella* in water systems include water temperature, stagnation, and the presence of free-living protozoa, which protect intracellular bacteria from adverse environmental conditions, including water disinfection procedures. Scale and organic sediment also provide nutrients for the formation of biofilms in which legionellae can persist for a long time, sometimes also for decades.

### 3.1. Water Temperature

The water temperature regime plays an important role in *Legionella* survival and growth. *Legionella pneumophila*, in particular, multiplies at temperatures between 25 °C and 42 °C, with an optimal growth temperature of 35 °C [2]. Thus, as indicated by several guidelines and technical reports, a key control measure of distribution systems colonization is to maintain elevated temperatures for the hot water and low temperatures for the cold water. The Centers for Disease Control and Prevention (CDC) guidelines for prevention of waterborne microbial contamination within the distribution system recommend to maintain hot water temperature at the return preferably at ≥51 °C, and maintain cold water temperature at <20 °C [3]. The European Technical Guidelines for the prevention, control, and investigation of infections caused by *Legionella* species [4] underline that avoiding water temperatures of between 20–25 °C and 50–55 °C in any part of a water system for any period of time is an important factor in controlling the risk. Finally, Italian Guidelines for the prevention and control of legionnaires’ disease [5] suggest a temperature of ≥60 °C at the water heater and a minimal water temperature of 50 °C across the network. However, in large buildings with complex plumbing systems, it can be difficult to reach these temperatures. On the other hand, elevated water temperatures accelerate disinfectant decay (e.g., chloramines and chlorine) [6,7] and predispose hot water systems to deteriorating microbial water quality [8]. Finally, when it is possible to achieve and maintain high water temperatures, the risk of scalding increases significantly. To minimize this risk, the already mentioned CDC guidelines suggest to install preset thermostatic valves (TMVs) in point-of-use fixtures which, in turn, may cause some drawbacks.

A TMV is a valve that blends hot water with cold water to ensure safe, constant water temperatures, typically between 38 °C and 46 °C, to water outlets. The TMV can be fitted between the hot and cold supply pipework and the outlet tap, or directly at the tap. The mixed water downstream of a TMV may provide an environment favorable to *Legionella* colonization, thus increasing the risk of exposure. Obviously, this needs to be managed. TMVs are very important for scald protection for anyone and particularly important for the very young, such as in schools and childcare facilities, and for the elderly and infirm, such as in care homes and hospitals. On the other hand, these settings—and in particular the hospitals and care homes—are of major concerns because they host persons particularly susceptible to the infection and at risk for developing severe clinical complications. Therefore, the use of TMVs should be decided by preliminary assessing the risk of scalding against the risk of infection from *Legionella*: If the risk of scalding is insignificant, TMVs are not required. Otherwise, where they are required, TMVs routine maintenance must be carried out by competent persons in accordance with the manufacturer’s instructions. Maintenance implies regular inspection, cleaning, descaling, and disinfection of any filters associated with each TMV. Finally, several cautions should be considered when TMVs are installed. First of all, they should be fitted as close as possible to the point of use in order to minimize the amount of stored blended water and, in turn, to avoid the risk of *Legionella* colonization. Secondly, a single TMV serving multiple outlets can increase this risk. For this reason, prior to its installation, the system parameters need to be checked against the working parameters of the TMV in order to decide how many outlets it could serve [9].

When it is not possible to maintain hot water temperatures above 50 °C and buildings cannot be retrofitted for TMVs, periodically increasing the hot water temperature to ≥66 °C at the point of use can be useful in the control of the colonization [3]. Anyway, although this type of thermal disinfection can be regarded as an effective control method, it has been seen that the genus *Legionella* has shown adaptation to a wide range of temperatures. Laboratory studies, for example, have shown that it can survive for 80–120 min at 50 °C and 2 min at 60 °C [10]. In real settings, some species, including *L. pneumophila*, were found able to grow at temperatures above 50 °C, and to survive up to 63 °C [11]. Moreover, in some circumstances the temperature range between 50–59 °C has been identified as the optimal condition for facilitating the emergence of different *Legionella* species [12]. Adaptations, spontaneous mutations or horizontal gene transfer from thermophilic *Legionella* species could support this eventuality. In particular, high levels of horizontal gene transfer responsible for the acquisition of thermotolerance have been shown for members of the genus *Legionella* within species and at the genus level by inter- and intra-species genome [13]. Furthermore, recently no significant correlation between distal site *Legionella* positivity and hot water return line temperature has been found. In particular, also buildings with a hot water return temperature >51.1 °C have been found positive for the bacterium in distal sites [14]. These findings imply not only that temperature should not be used alone to predict the presence or absence of *Legionella* in a drinking water distribution system, but also that it is crucial to evaluate the effectiveness of the applied thermal water management program.

### 3.2. Protozoa and Biofilm

Thermal conditions also influence the relationship between *Legionella* and protozoa. *Legionella* can enter free-living amoebas by phagocytosis and is generally killed in lysosomes, where an acidic pH and lysosomal enzymes digest the bacteria. However, *Legionella* has evolved temperature-dependent strategies to prevent lysosome-mediated destruction and persist in a vacuole within the protozoan hosts. In particular, at temperatures >25 °C, *L. pneumophila* replicates into amoebas and is released in the water while the protozoan host is killed. While at temperatures below 20 °C, amoebas infected by *L. pneumophila* eliminate the bacterium to the extracellular environment through a process of encystation [15].

Obviously, the need to avoid protozoan colonization of a water system is important not only to prevent the transmission of protozoa themselves [16] but also to decrease the risk of legionellosis because legionellae inside the protozoan cysts are protected against many harsh conditions. Thus, one of the key issues for controlling the growth of legionellae within protozoa is to recommend an effective disinfection method. However, there are several reasons why it is difficult to kill legionellae associated with protozoa.

When exposed to high temperatures or to chemicals, or in the absence of nutrients or oxygen, free-living protozoa transform to the cyst form to survive. It has been observed that the cysts can remain viable for more than 20 years under dry conditions [17] and 24 years at 4 °C in water [18], because they are metabolically inactive. The minimal temperature to kill the cysts is 65 °C for most *Acanthamoeba* sp. isolates [19], but the cysts of some thermotolerant isolates could resist exposure at 80 °C for 10 min [20]. Amoebal cysts have also shown to be highly resistant to some high-level disinfectants such as biguanides, quaternary ammonium compounds, chlorine, and hydrogen peroxide, as well as to UV, X-ray, and gamma irradiation [21].

Also, biofilm plays an important role in providing favorable conditions in which *Legionella* can grow because it provides protection from environmental stresses or disinfectants, access to higher levels of nutrients, and opportunities for symbiotic interactions with other microbes or protozoa. Interestingly, *Klebsiella pneumoniae*, *Flavobacterium* sp., *Empedobacter breve*, *Pseudomonas putida*, and *Pseudomonas fluorescens* are able to enhance the long-term persistence of *L. pneumophila* in biofilms [22,23,24,25] through the synthesis of capsular and extracellular matrix materials, which support the adherence [26,27,28], or because they provide growth factors for the bacterium [25]. On the contrary, other bacterial species such as *Pseudomonas aeruginosa*, *Aeromonas hydrophila*, *Burkholderia cepacia*, *Acidovorax* sp., and *Sphingomonas* sp. [25,29] antagonize the presence of *L. pneumophila* within the biofilm through the production of homoserine lactone quorum sensing (QS) molecules [30], or the production of bacteriocins [29]. It has been demonstrated that legionellae inside the biofilm matrix express phenotypes that differ from those of their planktonic counterparts and display an increased resistance to biocide treatments [31,32]. Thus, disinfectant type as well as substratum play an important role in the survival of *L. pneumophila* in biofilm within drinking water systems.

### 3.3. Organic and Inorganic Sediment Accumulation

Organic and inorganic compounds in water usually enhance *Legionella* premise plumbing colonization in the presence of ideal environmental factors, such as water temperature or pH, low water pressure or stagnation, loss of disinfectant, or discontinuous disinfection.

Limiting the levels of assimilable organic carbon (AOC) in distributed water could reduce the risk of bacterial re-growth [33], but for controlling *Legionella* it should be associated with other operational approaches [34]. In fact, proposed limits for AOC are related to the presence or not of a water treatment, ranging from 10 μgL^−1^ [35] to 32 μgL^−1^ [36] in non-disinfected water, and 100 μgL^−1^ in disinfected water [37]. Finally, small amounts of inorganic nutrients such as iron, zinc, and potassium also enhance the growth of *L. pneumophila* [38,39].

## 4. Disinfection Strategies

### 4.1. Oxidizing Agents

Chlorine is widely used for its strong oxidizing power for primary disinfection treatment of potable water. It reacts with a variety of bacterial cellular components [40] and is able to permeabilize the cytoplasmic membranes causing leakage of proteins and DNA damage [41]. In order to inactivate microbial contaminants or to prevent a possible microbial regrowth in water distribution systems, the World Health Organization (WHO) recommends 0.5 mgL^−1^ for free chlorine [42].

*L. pneumophila* has shown resistance to high levels of chlorine by the formation of biofilms [43], and pipes corrosion can be expected when the disinfectant is dosed in a water distribution system. In particular, high levels of residual chlorine are responsible for copper and iron pipes corrosion, while lead may not experience increased corrosion [44,45]. Maintaining the pH of the chlorinated water above 8 may counteract the corrosive effect of chlorine on copper and iron. Blends of poly- and orthophosphates can be used as corrosion inhibitors in order to create a passivating film on the pipe wall. However, the use of polyphosphates for corrosion control entails considerable uncertainty and risk because they may increase the leaching of pipe metal into the water [46].

Finally, the formation of potentially toxic and carcinogenic disinfection by-products (DBPs) has been identified in chlorinated drinking waters. The trihalomethanes (THMs) are likely to be the main chlorine DBPs. Because they usually occur together, total THMs are considered as a group, with an allowable concentration in drinking water ranging from 80 µgL^−1^ [47] to 100 µgL^−1^ [48]. THMs are formed by the interaction of aqueous-free chlorine with soluble, natural organic compounds present in water and their concentrations tend to increase with the presence of bromide or iodide, as well as with the increase of temperature, pH, and chlorine dosage [49]. Removal of THMs or their precursors is difficult and involves resource-intensive processes [50] like for example ozonation and biological granular activated carbon [51] or magnetic graphene oxide [52]. Nevertheless, the technologies described above could be time-consuming and extremely expensive. For this reason, an effective method to control DBPs in drinking water may be the use of alternative disinfectants.

Chlorine dioxide is a water-soluble gas that can easily diffuse through bacterial cell membranes. When compared to chlorine, chlorine dioxide has been found to be superior in penetrating biofilms [32] and in inactivating free-living protozoa such as *Acanthamoeba* strains [53].

It has been demonstrated that using chlorine dioxide as secondary disinfectant can significantly reduce the risk of acquiring Legionnaires’ disease in hospital settings [54,55,56]. A total chlorine dioxide residual of 0.1–0.5 mgL^−1^ at the tap is usually sufficient to control *Legionella* colonization, but higher residuals may be necessary in heavily colonized water systems [9].

Chlorine dioxide is less corrosive than chlorine [32]. Nevertheless, there is evidence that it can cause damage to polyethylene pipes [57,58]. Finally, although it does not form appreciable amounts of THMs and halo acetic acids (HAAs), it has been seen that on-site generation of chlorine dioxide often involves the production of free chlorine, which tend to increase with reaction time. Therefore, the ratio of chlorine dioxide to chlorine needs to be constantly monitored and possibly adjusted to obtain the best control of DBPs formation [59].

Monochloramine is formed by the reaction of ammonia with chlorine. For primary disinfection of drinking water, the target concentration for the disinfectant is 1.5–3.0 mgL^−1^ as Cl_2_ but the optimal concentrations may depend on the manufacturer. The WHO recommends 3 mgL^−1^ [60] while the Environmental Protection Agency (EPA) concentration is 4.0 mgL^−1^ as Cl_2_ [61]. However, monochloramine is often used as a secondary disinfectant after primary treatment with chlorine, especially in distribution systems heavily colonized by *Legionella* [62].

Monochloramine seems to react slowly with nucleic acids but rapidly with several amino acids, with little damage to bacterial membranes [63]. This could explain why measurements of free chlorine and monochloramine biofilm penetration show that monochloramine is more effective at penetrating biofilms than is free chlorine, although increased penetration does not correlate with greater inactivation of biofilm microorganisms. However, monochloramine has shown to be more effective on copper biofilms while free chlorine is more effective on polyvinyl chloride drinking water biofilms [64]. Finally, the lower reactivity of monochloramine can also be an advantage, as it is less likely to react with natural organic matter in the water, forming fewer DBPs, and leading to fewer undesirable tastes and odors than chlorine or chlorine dioxide [65].

The main disadvantage of monochlorination of the water is the nitrification by nitrifying bacteria of ammonia or free ammonia released by the decay of the disinfectant. Nitrification results in the formation of toxic nitrite in the distribution system. Moreover, the nitrification process has the potential to locally lower the pH in alkaline waters and cause corrosion of elastomeric materials. Factors influencing nitrification include the rate at which monochloramine residual decays, microbial regrowth, corrosion of pipe materials, as well as biofilm formation. Biofilm, in particular, facilitates the growth of nitrifying bacteria, promoting nitrification in water systems [66,67]. One way to potentially control nitrification in premise plumbing systems is through the development of specific building management plans. Optimizing the chlorine-to-ammonia ratio, typically 5:1 [68], flushing out the outlets [69], decreasing monochloramine residence time in service reservoirs [70], chlorite addition [71], reducing natural organic matter and preventing biofilm formation [34,72,73], as well as controlling the pH [74,75] may be effective methods of controlling and preventing nitrite formation.

Ozone is considered the most efficient disinfectant for all types of microorganisms because it is able to oxidize the cell membrane and wall constituents of a bacterial cell, as well as enzymes and nucleic acids. Under experimental conditions, it has demonstrated to be more effective at inactivating *L. pneumophila* than other disinfectants [76,77]. Nonetheless, determination of the in vitro activity of ozone against the bacterium does not predict the efficacy of its eradication from water fixtures [78]. In fact, studies in real water systems show no significant reduction in *Legionella* colonization [79,80].

Its low effectiveness against *Legionella* is probably due to the fact that ozone does not stay in water sufficiently long to provide a residual effect. For this reason, chlorine can be added after ozonation, but DBPs may be produced by their combinations. In fact, the vast majority of the ozone DBPs identified up to now contain oxygen in their structures, with no halogenated DBPs observed except when chlorine or chloramines are applied as a secondary disinfectant [81]. Moreover, there is evidence that the use of ozone as a primary disinfectant may cause a shift to more brominated DBPs during subsequent chlorination [82].

Hydrogen peroxide is a strong oxidizing agent that oxidizes microorganisms’ enzymatic systems, releasing free oxygen atoms without the formation of DBPs. The European directive established a concentration limit for hydrogen peroxide in drinking water of 25 mgL^−1^ [83]. While the US Environmental Protection Agency [84] guidelines recommend 25–50 ppm of residual hydrogen peroxide in drinking water. Concentrations of 25 mgL^−1^ have demonstrated good efficacy in controlling *Legionella* colonization of water networks, with a higher disinfection power with the increase of the water temperature up to 40–50 °C [85,86]. Finally, to enhance its activity, hydrogen peroxide is sometimes used in combination with other oxidants such as ozone, silver, or UV irradiation.

### 4.2. Non-Oxidizing Agents

Copper–silver ionization is commonly used in water distribution systems with hot water recirculating loops [32]. When used together, copper and silver ions create a synergistic effect. In fact, the copper ions destroy the cell wall permeability, which allows the silver ions to interfere with the synthesis of proteins and enzymes, thus resulting in a higher inactivation rate of *L. pneumophila* [87]. For *Legionella* control, the recommended copper and silver ion concentrations are 0.2–0.4 mgL^−1^ and 0.02–0.04 mgL^−1^, respectively [32].

Literature data show that there are at least three major challenges facing copper–silver ionization [88,89,90]: (*i*) ensuring that the added ions are flushed throughout the entire water distribution system, (*ii*) the emergence of resistant legionellae, and (*iii*) the reduction in the microbiocidal power due to the formation of metal complexes. Finally, copper–silver ionization may increase chlorine DBPs formation at pH 8.6 in the presence of natural organic matter [91].

Short-wavelength UV is believed to have biocidal effects through a molecular rearrangement of the purine and pyrimidine components of the nucleoproteins, which hampers DNA replication [92]. Breaks in the bonding structure also occur [93]. Nevertheless, the current technology of low-pressure (LP) mercury lamps that is used for UV irradiation has several shortcomings.

First of all, *Legionella* is able to repair damages to DNA [94]. Secondly, the maximum absorbance of nucleic acids is around 260 nm, while proteins have a relative maximum absorbance of around 280 nm [95]. Thus, targeting one of the components of the bacterial cell may be more effective in *Legionella* inactivation [96]. LP mercury lamps emit at 254 nm, and this may result in a low efficacy of inactivation. On the contrary, the irradiation technology using light emitting diodes (LEDs) can emit at many different wavelengths in the UV-B and UV-C. Moreover, LP lamps contain mercury, which is toxic to the environment, while LEDs are made of gallium/aluminum nitride or aluminum nitride, which are not toxic nor hazardous.

Regardless its technology, UV light produces no residual. Therefore, if a residual is desired, another disinfectant must be used. To this regard, the application of UV light to shower heads and faucets in hospitals has indicated that the technology alone is insufficient to control *Legionella*, and other disinfection measures have to be used along with UV irradiation for an effective control of the colonization [92]. Finally, UV light shows poor penetration in biofilms [97].

### 4.3. Point-of-Use (POU) Filtration

Disinfectants may not reach dead legs or low water flow areas. Furthermore, they can dissipate through the plumbing system, making them less effective at distal points, and secondary chemical water disinfection may shorten the life of the pipes. For these reasons, point-of-use (POU) filtration can be regarded as an additional barrier to add to other primary treatment technologies to prevent exposure to legionellae, particularly in hospital settings. The filters are attached to individual faucets and showers, providing a physical barrier between *Legionella* and high-risk patients. Anyway, a recommendation for their use cannot be made until an evaluation of their efficacy has been performed [98].

It has been seen, for example, that the application of carbon filters may result in a greater presence of *Legionella* in water because the bacterium can colonize the filters while passing through [99,100]. On the contrary, membrane filters have shown to control the colonization of hospital water systems up to 2–8 weeks of continuous use [101,102], especially when they are covered with a silver layer [103]. More recently, the novel electrically heatable carbon nanotube (CNT) point-of-use (POU) filters have demonstrated to remove 99.9% of *L. pneumophila* in water [104].

Obviously, one filter is needed for each fixture, and if the filters are not regularly replaced, *Legionella* can colonize them regardless of their technology. Therefore, replacing these filters may be expensive. For this reason, the use of filters, which requires fewer change-outs, could be a cost-effective method for preventing hospital-acquired Legionnaires’ disease [105].

## 5. Emerging Treatment Technologies

As discussed above, chemicals dosed into the water systems could have a negative impact on the environment and human safety. For these reasons, alternative disinfection strategies are needed. Literature reports some alternative treatment technologies against legionellae that are, mostly, based on natural biocide molecules, or structural and/or chemical modification of materials to be used as repellant surfaces for water pipes and filters.

### 5.1. Essential Oils (Eos)

Essential oils (Eos) are aromatic oily liquids obtained from plant material (e.g., flowers, seeds, leaves, or roots). They are mainly composed of a mixture of terpenoids and aromatic compounds, which have shown an antibacterial activity. Although EOs antibacterial activity could not be linked to one specific mechanism, scanning electron microscopy analysis has shown morphological alterations of *L. pneumophila* when treated with *Thymus vulgaris* EO, suggesting a loss of cytoplasmic membrane integrity [106].

EOs extracted from *Cinnamomum osmophloeum* leaves, in particular, have demonstrated a strong activity against *L. pneumophila* at 42 °C in cooling towers [107] and at pH 8–10 in recreational spring waters [108]. Lemon, peppermint, sage, and thyme oils have also shown to have a good anti-biofilm activity [109]. Finally, *Melaleuca alternifolia Cheel* (tea tree) oil (TTO) has demonstrated to be active as anti-*Legionella* disinfectant for control contamination especially in spas, in small water distribution systems or in respiratory medical devices [110].

### 5.2. Biosurfactants

Biosurfactants are a group of natural biocide molecules produced by bacteria, yeasts or fungi, which seem to act by direct lysis of negatively charged membranes [111]. So far, only few studies have reported the anti-*Legionella* activity of biosurfactants produced by *Pseudomonas* strains [112], and various *Bacillus* environmental isolates [113]. Surfactins produced by *Bacillus subtilis*, in particular, were found to breakdown *L. pneumophila* pre-formed biofilms, but they did not prevent biofilm attachment [114]. Finally, cell-free supernatants produced by *Lactobacillus rhamnosus* and *Lactobacillus salivarius* showed high antibacterial activity against *L. pneumophila* strains isolated from hot tap water [115].

### 5.3. Smart Surfaces

Smart surfaces are an emerging material technology able to respond dynamically to changes in the environment via the incorporation of stimuli-responsive polymers (SRPs) into the material surface. Various stimuli may be used to trigger antibacterial action [116]. A smart surface composed by nanostructured silicon nanowires has been recently proposed. Once chemically modified with (3-aminopropyl)triethoxysilane and chlorhexidine digluconate, these nanowires seem to provide an efficient decrease of planktonic and surface-attached microorganisms, thus reducing their subsequent survival and the possibility of adaptive process [117].

A sulfonated pentablock copolymer (s-PBC, commercially named Nexar™) has demonstrated to induce the death of *P. aeruginosa* by a contact killing mechanism in which the polymer acidifies the water close to its surface, causing the microbial replication inhibition [118]. According to the reported previous work, the same authors performed a preliminary experiment using *L. pneumophila* sg 2-14 instead of *P. aeruginosa*. A modified Zone of Inhibition test using polypropylene (PP) coupons that have been deposited with the s-PBC (s-PBC@PP) was used to test its bactericidal activity. In this case, 0.2 mL of water were spotted below a s-PBC@PP and a reference PP coupon in a Glycine Vancomycin Polymyxin Cycloheximide (GVPC) Legionella agar selective medium inoculated with the target bacteria. As reported in Figure 2, after 24 h a clear zone of 2 cm diameter appeared all around the s-PBC@PP coupon (A), while microbial colonies were still visible nearby the PP coupon (B), proving that the s-PBC, acidifying the water volume, was able to induce the death of those legionellae that were directly exposed to the modified surface. The results of this first test are promising, and further investigations are in progress.

## 6. Water Sampling

As underlined by many guidelines and technical reports, one of the main objectives of water sampling for environmental surveillance is confirmation or exclusion of a water system as a source of infection. Routine environmental sampling could also be useful for distinguishing between local or system-wide colonization of the water system, as well as for identifying critical sites and selecting the appropriate strategy for short- and long-term control of *Legionella*.

The number and types of sampling sites must be determined on the water system basis because of the diversity of the environmental conditions, namely water physical and chemical characteristics, plumbing materials, heating, disinfection method, etc. Key components of water sampling also include site selection and sampling frequency, sample type (i.e., biofilm, water, etc.), as well as flow patterns (e.g., first-flush, post flush).

For a full system perspective, water samples should always be taken from water mains, water softeners, holding tanks, water heaters (inflows and outflows), faucets, and shower heads at the proximal and distal end of the water system and at a number of representative points in between. Selection of sampling sites depends on whether the sampling is for routine monitoring or to investigate an outbreak.

For routine monitoring, sampling points are chosen taking into consideration the sectors of the building more at risk of *Legionella* proliferation (e.g., dead-ends or areas with infrequent use, etc.). Furthermore, monitoring the temperature of cold and/or hot water systems (e.g., the hottest and the coldest point, respectively) could be a useful approach for identifying points that are susceptible to *Legionella* colonization [119].

In the case of outbreaks, sampling plans are useful to identify all potential sources of contamination. For cold water, the main sampling points are the incoming supply, the outlet of each reservoir, as well as outlets closest to the reservoir. Sampling of hot water systems should include showers and taps used by infected people or in proximal areas. Finally, evaporative cooling systems, spa pools, humidifiers and decorative fountains should be considered as possible sources of infected aerosols. [120].

Researches on environmental monitoring show that testing water regularly for the presence of *Legionella* may also be a guide to verify the efficacy of disinfection. The WHO, in particular, recommends drinking water cultures for *Legionella* every 3 months [121].

In Sicily (Italy), surveys of hospitalized and non-hospitalized (marine, touristic and corporate) facilities in the central-eastern part of the region revealed ~ 46% and 30% of hot water systems highly colonized by *Legionella*, respectively, as reported in Table 1. *L. pneumophila* 2-16 was the most frequently isolated serogroup [62,122,123,124,125,126]. For each facility, a first preliminary inspection of the water system conditions focusing on cleanliness, water turbidity, wall slime, sludge, general repair, access, and location of the sampling points was carried out. Furthermore, technical information regarding the plumbing systems (e.g., pipe materials), water treatment (e.g., type and concentration of the disinfectant, continuous or discontinuous disinfection), previous notification of cases of legionellosis etc., were collected. In hot water systems, the following sampling points were always considered: water heaters set at low temperatures (<50 °C), water tanks, water softeners, taps, and faucets (pre-flush, post-flush).

The role of routine testing of water systems for the monitoring of *Legionella* has been subjected to some scientific debate. Sometimes it has been shown the lack of correlation between water sampling results and human health risk [127,128]. This may be due to the fact that each sampling event might not capture the true dynamics of *Legionella* within a water system. In fact, a study examining 84 taps over 5 consecutive days in an Italian hospital revealed significant variation of the *Legionella* load from day to day, although the pattern was similar across the wards monitored [129]. Additionally, it has been shown that high percentages of legionellae in water systems are not culturable.

## 7. Viable but Non Culturable (VBNC) State

Legionellae are able to persist in adverse environmental conditions as vegetative cells with low metabolic activity, via the activation of the viable but non culturable (VBNC) state. VBNC cells are viable cells that have lost their ability to grow on routine culture media but they generally maintain their ability to cause infection [130,131]. It has been recently demonstrated that VBNC legionellae can directly infect human macrophages and amoebae even after 1 year of starvation in ultrapure water. Thus, amoeba infection by starved VBNC strains, which usually occur in biofilms, will result in an increase of legionellae in water systems [132]. The real number of metabolically active legionellae will be, in turn, underestimated by the use of conventional culture techniques.

It has been seen that under stressful conditions (e.g., chemical disinfectants, UV light, heat, etc.), two different types of VBNC legionellae can be produced [131]: (i) the damaged VBNC type, in which the cells are on the way to death and thus they have lost their ability to infect humans; (ii) the injured VBNC type, in which legionellae retain their pathogenicity because they undergo a physiological adaptation to stress through a change in the gene expression and may resuscitate when the stress is removed. Therefore, it is of paramount importance to apply the most reliable detection method able to differentiate between culturable and VBNC cells.

### 7.1. VBNC-Legionella Isolation Methods

The gold standard method for *Legionella* detection is the routine culture technique, consisting in water filtering and plate counts of visible colonies [133]. However, direct colonies observation is not the best solution for the VBNCs identification and quantification, since it tends to overlook those cells that are viable, potentially proliferating and virulent, but non-culturable [134]. In this sense, many efforts have been made in order to improve the performances of culturing methods. Some attempts, for example, were based on the integration of cytometry to reduce the time consumption. Other trials were based on the addition of a preliminary immune-separation step to increase the yield by identifying the total number of viable cells (including the not culturable) [135,136]. Some efforts were, instead, focused on the resuscitation of VBNCs by co-culture with amoebae to make *Legionella* cells again culturable and restoring their colony formation and identification, but implied a long procedure resulting in being unsuitable for the routine analysis of VBNCs [137].

To solve the ambiguity issue in the VBNC detection, other approaches focus on the VBNCs identification and counts via the bacteria membrane staining. These methods are based on the analysis of membrane integrity (in terms of protection, nutrients transport, oxidative phosphorylation, etc.) by labelling bacteria with specific hydrophilic dyes such as the propidium iodide. In this way, the intact phospholipidic bilayer of live cells blocks the dye penetration, while dead cells with damaged membranes allow the chromophore to pass and bind the DNA, giving a red-orange fluorescence after excitation with a 488 nm light [138]. Many kits (e.g., BacLight kit from Thermo Fisher) use this kind of dye in combination with others able to penetrate the healthy bacterial cells, giving a dualistic fluorescent response [139]. However, these kits have the main limitation of live cells number overestimation, since not all the dead cells have their cell membrane totally and/or always compromised.

Another cell parameter used to evaluate the cell viability and the VBNC number is the membrane potential. Some evidences report methods based on fluorescent probes that can pass only those membranes having a zero potential to stain only those cells that are in starvation condition (i.e., the VBNCs) [140]. Other membrane potential based methods report the use of cationic probes (e.g., potassium ions) that, after a prolonged exposure to measured bacteria, are spectroscopically quantified inside cells in order to test the decrease of the transmembrane transport activity [141]. This feature was also evaluated analysing the uptake of redox- or radioactive probes by electrochemical or micro-autoradiography analysis [142]. Sometimes, instead, the membrane potential analysis is combined with flow cytometry to measure the change of light scattering given by the size decrease of VBNC cells, allowing their quantification by count of the dimensional variants. However, for most of the proposed approaches, there is a high percentage of positive signals that can be obtained with viable and culturable cells [143], which makes the membrane potential-based methods not totally exhaustive for an accurate VBNCs detection and quantification.

The metabolic analysis of *Legionella* solves some of the previous issues and drawbacks, focusing on the functional characterization of enzymes that are essential for the bacteria physiology preservation, such as esterase, arylamidase, phosphatase, phosphohydrolase, and glucosidase [144]. The esterase activity, for example, can be tested by using specific substrates that act as membrane permeant fluorescent dyes. These molecules reveal their fluorescence only if esterase is active, since the enzyme cleaves the substrate giving a fluorescent derivate (and a specific light emission), and if the membrane is totally intact, allowing the fluorescent product to be retained and accumulated [145].

The fluorescence in situ hybridization (or FISH) is, also, reported to be a valid technique for the VBNC detection. Based on the direct intracellular hybridization of labelled oligonucleotide probes, as the 16S rRNA [146], this approach provides a viability assay through the bacteria transcriptome analysis, revealing the expression of vital genes. However, it is often affected by weak probes fluorescence signals and/or low detection specificity, caused by the cross-reactivity towards other *Legionella* species [145]. Some evidences, however, reported improved FISH approaches as the combined Direct Viable Count-FISH (DVC-FISH), a rapid and specific tool to identify pathogenic *Legionella* spp. and *L. penumophila* viable cells harboured by amebae in water sources [147], and the Catalysed Reporter Deposition-FISH (CARD-FISH), an enhanced FISH to detect cells transcriptome with low ribosome content [145], which excludes the risk of cross-reactivity. However, some scenarios, such as those caused by the heat or chemical treatment, report intact rRNA molecules inside cells that have compromised their membranes, so that the rRNA analysis by FISH itself results to be not totally reliable as a proof of Legionella viability.

Molecular techniques, mostly based on the quantitative real-time PCR (qPCR), are valid alternative approaches able to guarantee a rapid, sensitive, and specific detection of *Legionella* and its VBNC [148]. Conventional qPCR methods, like for example those based on the use of 16S rRNA as viability indicator [149], can perform an accurate detection of *Legionella* in a few hours although they are limited by a DNA persistence in dead cells, which causes an overestimation of viable cells and a risk to collect false-positive results [150]. Thus, new PCR approaches combining the amplification reaction to some permeant nucleic-acid-binding dyes have been introduced. These viability PCR (v-PCR) methods provide an exhaustive discrimination between live and dead cells thanks to dyes such as the propidium monoazide (PMA) and ethidium monoazide (EMA) that are able to penetrate only damaged membranes of dead cells and bind covalently to the intracellular genome, thus blocking its subsequent amplification by PCR. In this way, the amplification occurs only for viable cells, reducing the risk of VBNCs discarding and dead cells overestimation, which is typical of conventional qPCR and culture methods [148,151]. Moreover, some evidences of combined v-PCR approaches, such as that including a preliminary immunomagnetic separation step before the PMA-PCR reaction to increase the level of cells purification [152], suggest the possibility to improve these amplification methods, providing sensitive, selective, simple, and rapid bacteria detections in water towards innovative environmental monitoring applications.

### 7.2. Emerging Detection Methods

Nowadays, many research efforts have been spent in the development of advanced biosensors for the viable bacteria revelation in water. Biosensors are versatile and customizable systems able to guarantee a biological analysis without the need for bulky instruments and complex protocols, thus reducing the lab constrain and time/cost consuming typical of consolidated methods. They basically work through a perfect synergy between the selective biological or biomimetic recognition element and the transducer (e.g., optical, electrical, electrochemical, etc.) operating the target revelation [153,154]. If opportunely implemented, biosensors can provide a fast revelation of bacteria together with a high degree of miniaturization and integration of the entire analytical system, ideal for in situ and remote-controlled applications. The antimicrobial peptide (AMP)-based biosensors, for example, introduced rapid, reliable, and cost-efficient tools for the direct recognition of floating bacteria in contaminated water [155]. The “watersampling” chip has shown to be able to reveal viable pathogens in water by the real-time electrical detection of their interactions with anchored AMP, reaching a Limit of detection (LoD) of 1 cell/µL [156], which is promising towards the VBNCs detection.

Another example is represented by the silicon-based biosensors, which have brought an important breakthrough in the biological sensing thanks to the properties of the material used. Silicon, in fact, is a biocompatible substrate that allows to embed all the stuff required for the sensing process, from probes and reagents to the detection module, and is suitable for a wide series of further implementations, as dedicated staining techniques [157,158], microfluidics integration [159], detector downscaling (integrated microelectrodes, micro-sized photomultipliers, etc.) [160,161], and nanostructured-based functionalization [162]. This implementation is reported, for example, by authors who recently developed an optical system based on a nanostructured oxidized porous silicon (PSi) thin film exposing specific antibodies against *Escherichia coli* for its real-time detection in water. The system, reporting a LoD of 10^3^ cells/mL, works through an optical revelation of the bacteria-antibodies complex performed by directly measuring their light reflection through a charge-coupled device (CCD), an integrated circuit containing an array of linked or coupled capacitors [163]. Most of the biosensors described above have been validated only for few pathogens species such as *E. coli* and *Salmonella* [164] but considering the analytical approaches and performances reported, they could be reasonably applied to the VBNC-Legionella detection.

## 8. Water Systems Management

Water systems management programs are recommended to prevent *Legionella* growth in buildings with large or complex water systems, including health care facilities. Understanding the biological and environmental factors that contribute to the persistence and colonization of a water system can be detrimental to eradicate and prevent the transmission of *Legionella*.

As discussed above, although surveillance of water systems can be regarded as an important component of the primary prevention of legionellosis, especially in hospital settings, risk assessment based solely on counts of *Legionella* may be misleading and financially consuming [127,128]. Instead, information about the presence and the extent of the colonization of the water system should be spent on managing appropriate control measures within a Water Safety Plan (WSP) [165].

### Water Safety Plan (WSP)

Water Safety Plans (WSPs) are drawn on the principles from other risk management strategies, such as the multiple-barrier approach and Hazard Analysis and Critical Control Points (HACCP). According to the indications of the WHO [165], a WSP is composed of four main steps.

The first step, “system assessment”, provides a systematic assessment and prioritization of hazards and control measures related to the water system. The size and complexity (e.g., pipe materials, dead legs, water flow conditions, etc.) of the water system might increase the risk of *Legionella* colonization. Thus, a comprehensive understanding and description of the system is fundamental for the identification of the hazards and the assessment of the risks. For this reason, the first stage in developing a WSP is to form a multidisciplinary team of professionals with sufficient expertise in building management, as well as in distribution and treatment of drinking-water, such as engineers, water quality specialists, hygienist professionals, and operational staff.

A semiquantitative matrix is typically used for identifying the likelihood of occurrence of a risk (rare, possible, certain) and evaluating the severity of consequences of the hazard (insignificant, major, catastrophic): a risk ranked above the “cut-off” point will require immediate attention. Risk ranking will also provide a starting point for prioritizing remediation actions and will help the decision-making process.

Preventing *Legionella* colonization of the distribution system will depend on the design and operation of the system, as well as on the most effective maintenance and survey procedures. For this reason, the second step, “monitoring”, of a WSP focuses on the identification and evaluation of the applied control measures.

Typically, control measures are applied collectively to eliminate or significantly reduce the occurrence of an hazard. Identification and implementation of control measures require a multiple-barrier approach: a failure of one barrier may be compensated by the remaining barriers. For this reason, assessment of existing control measures is necessary in evaluating whether they are effective in reducing the risk to acceptable levels. Obviously, if improvement is required, alternative and additional control measures that could be applied should be evaluated.

As already discussed, *Legionella* grows better in water systems that are not adequately managed. In general, the principles of an effective water management include measures that prevent *Legionella* growth, such as assuring water temperatures outside the ideal range; avoiding water stagnation; scale and corrosion, which in turn promote biofilm growth; as well as ensuring an adequate disinfection method [166]. A number of control measures can be adopted to reduce the risk of *Legionella* colonization of a water system. Taking into consideration the multiple-barrier approach, by decreasing the contamination of the source water or reducing retention of water in storage tanks, for example, the amount of chemical treatment required is reduced. This may in turn reduce the production of disinfection by-products (DBPs) and minimize operational costs. Other barriers to contamination of the water system may be those related to water disinfection. Application of an adequate concentration of disinfectant is an essential element to reduce the risk. Anyway, hazards may be introduced during treatment. Thus, where chemical disinfection is used, for example, measures to minimize DBPs formation should be taken into consideration. In this case, storage of water after disinfection and before supply to consumers could reduce chemicals’ demand and improve remediation by increasing disinfectant contact times. Control measures may also include using a more stable secondary disinfectant (e.g., monochloramine instead of chlorine). This can be particularly important when *Legionella* is associated with protozoa and/or biofilm and to prevent corrosion of pipe materials and the formation of deposits. Furthermore, storage tanks and reservoirs should be securely roofed and equipped with an external drainage to prevent *Legionella* contamination and to avoid water stagnation, respectively. Flushing and maintaining positive pressure throughout the entire water system could also be an effective control strategy to avoid stagnation. In fact, in systems where there are dead-legs or water is supplied intermittently, the risk of colonization is very high. Obviously, where household storage is used to overcome intermittent supply, localized use of disinfectants to reduce microbial proliferation may be necessary.

The third step, “management and communication”, describes actions to be taken during normal operation and in case of incidents and emergency situations (e.g., remedial actions after adverse monitoring results, notification of a case of legionellosis, etc.). This step underlines the need of proactive water management plans.

The management plan will include both responses to normal variations of monitoring parameters and responses when operational monitoring parameters reach or overcome critical limits. During an emergency, for example, actions may include restricting access to water or temporarily use an alternative water source, if possible. It may be necessary to increase the level of disinfectant or to additionally disinfect the water. In addition to the periodic review, it is important that the WSP is reviewed following every incident or unforeseen event in order to identify areas for improvement, revise risk for the risk assessment, or identify new risks, focusing on and acting against the cause of the incident.

Communication and collaboration among environmental and Public Health experts is also imperative. Communication strategies should include procedures for promptly advising any significant incidents within the water system, including notification of the public health authority.

The fourth step, “surveillance”, provides an indication of the overall performance of the WSP through the systematic collection of data (e.g., periodically sampling for legionellae in water, verifying the disinfectant level or water temperature) to verify if control measures applied are operating properly.

Although WSPs require a certain minimum number of steps involved, it should be regarded as a flexible approach that should build on and improve existing practice. Starting with existing treatments to ensure that they are operating at their optimum is one of the main key components of WSPs implementation, because this is often the principle barrier that prevents *Legionella* from colonizing the water distribution system. On the other hand, for reducing *Legionella* risks in new buildings, water systems should be designed to ensure and facilitate safe system operation and maintenance. For example, water systems should be designed to prevent microbial growth, minimize corrosion, and maintain internal surfaces in a clean condition. This designing should be regarded as an integral component of an effective risk control strategy. In particular, the following requirements for water systems should be assessed in the planning stages: avoiding water storage tanks supplying calorifiers; using point-of-use water heaters rather than centralized hot water systems; minimizing the distance between the source of the water supply and point of use; designing distribution systems to ensure regular circulation of water by eliminating dead legs; insulation of the pipes to minimize the transfer of heat between hot and cold water distribution pipework; installing thermostatic mixing valves (TMVs), when used, as close as possible to the point of use. Finally, as already discussed, legionellae are usually associated with biofilms in water systems. Thus, consideration should be given to the materials used in the construction of water distribution systems. Copper seems to be the best at limiting biofilm formation, followed by polybutylene and stainless steel, whereas polyethylene, chlorinated polyvinyl chloride (PVCc), unplasticised polyvinyl chloride (PVCu), steel, and ethylene-propylene encourage biofilm formation.

## 9. Conclusions

Growth of *Legionella* in water systems poses an increasing Public Health concern. Temperature, water quality, pH fluctuations, scale, changes in water pressure, stagnant water, pipe materials, as well as inadequate levels of disinfectant can influence the colonization and growth rate of *Legionella*. Biofilm also plays an important role in providing favorable conditions in which *Legionella* can grow by providing protection from the effects of biocides.

A preventive proactive approach should be based on specific water management programs, which might comprise the identification and assessment of the risk, the implementation of a WSP, and the application of the most appropriate disinfection treatment.

Routine water samplings to detect *Legionella* is necessary to assess the risk because the level of colonization may vary over time. Water samplings could be useful in monitoring the effectiveness of the control measures applied. However, the implementation of a risk assessment plan to predict the risk of legionellosis based solely on culture-based enumeration in a water system contains many uncertainties. Evaluation of factors that may indicate increased risk, such as population risk factors and/or presence of water system’s elements associated with an increased risk, could be more useful in establishing engineering control strategies. The Public Health departments might serve as a resource to building managers during the development, implementation, and evaluation of the water system management.

## Figures and Tables

**Figure 1 microorganisms-09-00577-f001:**
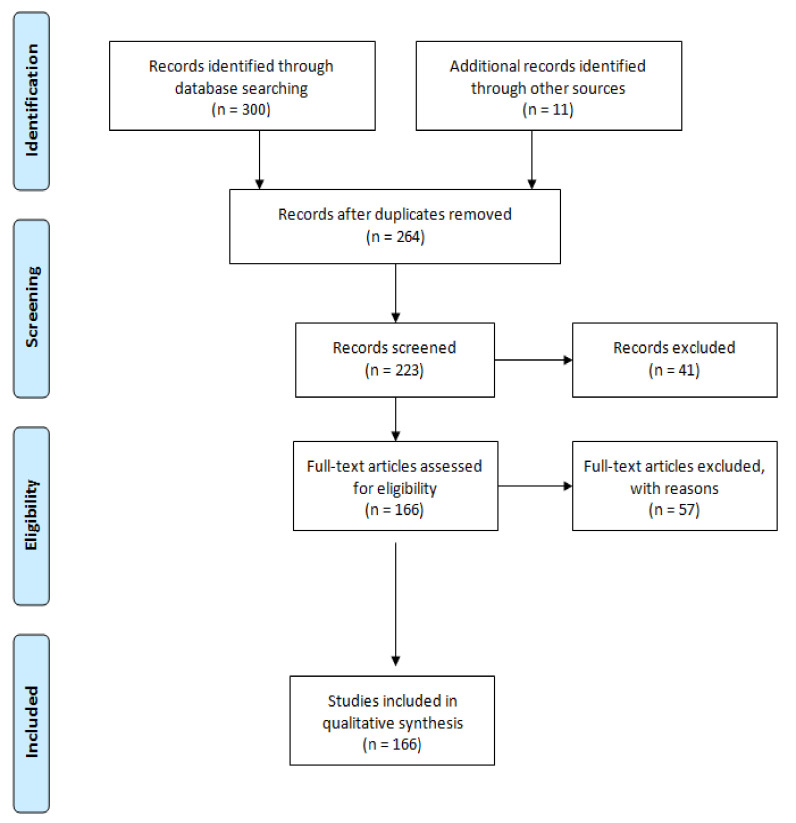
Flow diagram.

**Figure 2 microorganisms-09-00577-f002:**
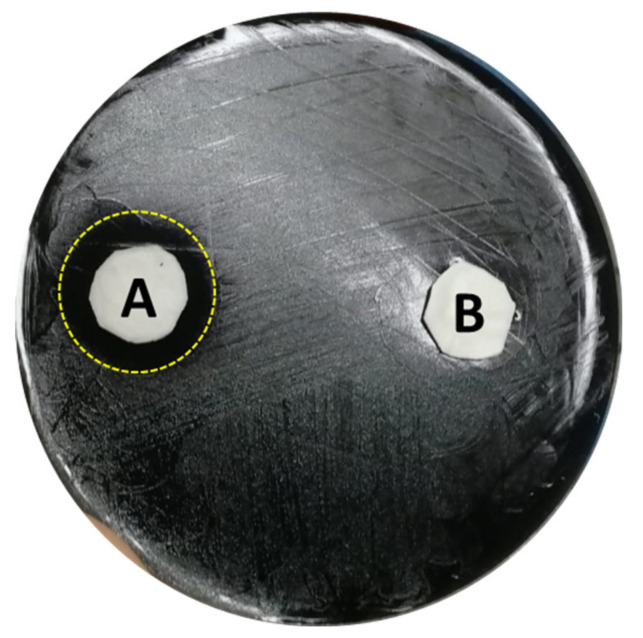
Modified Zone of Inhibition Test of *Legionella pneumophila* 2-14 after 24 h incubation with s-PBC@PP (A) and reference PP (B) coupons after 0.2 mL spotting of water. The yellow dot ring evidences the 2 cm Ø clear zone.

**Table 1 microorganisms-09-00577-t001:** *Legionella* screening in water systems of central eastern Sicily facilities.

Type of Facility	Total Samples	Positives (%)	SG 2-16 (%)
Hospitals	2610	1193 (45.7)	823 (69)
Marine	1133	246 (21.7)	184 (74.8)
Touristic	582	146 (25.2)	83 (56.8)
Corporate	196	69 (35.2)	61 (88.4)

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
