# Peer review of "Environmental Management of Legionella in Domestic Water Systems: Consolidated and Innovative Approaches for Disinfection Methods and Risk Assessment"

_microorganisms, 2021, doi:10.3390/microorganisms9030577_

Round 1

Reviewer 1 Report

The authors present an interesting study regarding the Environmental management of Legionella in domestic water systems and the application of WSP on this. It is obvious that the authors have made a remarkable effort to identify the relevant information and combine it aiming to prove ‘that the implementation of water management plans is the main prevention measure against Legionella’. 
However, while there is an extensive review on the risk factors for the abundance of Legionella and the disinfection procedures, a small section of the ms is devoted to the application of the WSPs and how these can prevent effectively the presence and growth of Legionella. I think that this section must be developed while some others must be merged (eg the disinfection methods). 
Additionally, the article does not follow the Journal’s instruction, which is very important when someone wants to publish a Review article! The article should conform to the PRISMA guidelines. It is very important to present the Methodology (the whole methodology section is missing). For example, which and how many keywords were used, which data bases were searched, how many articles were initially identified and how many articles were finally considered for further discussion in the current review. I would like to kindly ask from the authors to provide this information in a detailed methodology section which is very important in order to highlight the importance of their work.
The language needs to be revised throughout the manuscript preferably by a native speaker including the abstract. For example, please replace the word ‘anyway’ (which is mentioned many times in the ms) with another one more suitable. 
The study can be considered for publication if the authors are willing to clarify the above points and to add some infos in order to reveal the significance of their work.

Reviewer 2 Report

This referee consider the manuscript as an excellent review on the subject, well organized and understandable. I would just like to point out that the scientific names of microorganisms must be spelled correctly. The first time a species is written it must be completed. It doesn't matter if the genera wer already mentioned. The correct use of units of measurement should also be reviewed.

Round 2

Reviewer 1 Report

I would like to than the authors for replying to the initial comments. The article can now be considered for publication.